# Viability Study of the Application of Bi-Block Concrete Sleepers as a Solution for Technical Landfills

Camilo Másmela, Elisabete Teixeira *, Joaquim Tinoco, José Campos e Matos and Ricardo Mateus

Department of Civil Engineering, Institute for Sustainability and Innovation in Structural Engineering (ISISE), University of Minho, 4800-058 Guimarães, Portugal; pg41009@alunos.uminho.pt (C.M.); jtinoco@civil.uminho.pt (J.T.); jmatos@civil.uminho.pt (J.C.e.M.); ricardomateus@civil.uminho.pt (R.M.)
* Correspondence: b8416@civil.uminho.pt

**Featured Application: This work can be used to support decisions made by transportation infrastructure managers for circular development, using waste as raw materials.**

**Abstract:** Transition zones, specifically embankment structures on railway tracks, are recurrently damaged by high-speed rail traffic, thus producing abrupt variations in the track vertical stiffness. The main objective of this work is to study the implementation of Construction and Demolition (C&D) wastes, specifically Bi-Block (BB) concrete sleepers, to minimize the issues related to the differences in stiffness along the area of the technical landfills, using unconventional environmental solutions. The use of BB wastes can reduce the environmental impacts caused by the construction sector. The studied solutions produce similar stiffness levels to traditional solutions. The Multicriteria Decision Support Methodology for the Relative Sustainability Assessment of Building Technologies (MARS-SC) allows concluding that the studied solutions are more sustainable than the traditional ones.

**Keywords:** bi-block concrete sleepers; life cycle assessment; numerical modeling; railway; technical landfills; transition zones

## 1. Introduction

Construction sector activities, of which the railway system is a part, produce high volumes of waste, which has environmental implications associated with the consumption of natural resources, soil, greenhouse gas emission, and contamination of soil and aquifers [1]. Due to these implications, those responsible for the environment and construction sector have made significant efforts to create sustainable alternatives and reduce the consumption of extracted natural resources [2].

Worldwide, it is estimated that one-third of 4.6 thousand tons of Construction and Demolition (C&D) waste produced annually corresponds to a concrete element, of which only 5% to 10% is recyclable. The European Union (EU) generates more than 500 million tons of C&D waste annually. This waste stream represents between 25% and 30% of all waste produced [3]. Portugal has an ambitious plan for its expansion, repair, and rehabilitation in the framework of railway infrastructures. As a result, a high volume of C&D waste will be produced in the coming years. According to APA [4], 2.17 million tons of C&D waste were produced in 2018, and 1.7 million tons were destined for local duly licensed for effect, i.e., a valuation rate of 78.24%. LER normative rules specify common treatment options such as sorting, crushing, and screening before being implemented [5]. Currently, concrete bi-block (BB) sleepers that do not satisfy technical requirements represent a significant C&D concrete waste. It is not always easy to find a viable reuse solution, and there is an urgent need to create management solutions for the waste obtained from the maintenance of railways to reduce the level of environmental and economic impacts.

Currently, in the transition zones of railway tracks located between embankments and structures, there is higher stiffness of the structures, e.g., the bridges, the viaducts, the

culverts, the tunnels, the underpasses, and lower in the embankment zones [6,7]. High degradation rates are often observed there, which are attributed to abrupt variations in vertical stiffness at short distances subjected to dynamic loads, differential settlements, energy dissipation conditions, inadequate compaction, different support conditions, displacements due to temperature, and bridge deck displacements and rotations [8]. Different settlements can also occur due to the plastic behavior in the foundation layers when they are stressed with cyclic loads from trains in transit or consolidation is present in the track substructure. As a result, an unevenness in the track is produced, allowing the amplification of dynamic loads and causing severe damage to the railroad [9]. These stiffness variations can cause significant problems, such as accelerated track degradation, disruptions in railway operation, vertical rail unevenness, and differential settlements. Consequently, they require rigorous monitoring procedures and more frequent maintenance actions [10].

Therefore, traditional solutions were studied to mitigate and solve the problem resulting from the transition from the embankment to the engineering structures, e.g., transition wedges, and were contrasted with new solutions capable of giving similar results besides adding value due to the incorporation of elements catalogued as C&D waste (e.g., concrete BB sleepers) [11]. Nowadays, diverse standards have been developed for C&D waste, obtained from transport infrastructure, in a sustainable way [12–15].

Further, to increase a circular economy, it is essential to find new solutions for reusing, recycling, or transforming the waste discarded at construction sites or by demolition, especially those related to concrete [16], in this case study applied to BB sleepers. A Life Cycle Assessment (LCA) [17–19] was applied in this study to evaluate the environmental performance of various provisions when implementing BB sleepers as reinforcement in transition zones. Thus, it was intended to carry out a feasibility study of the application of BB sleepers as C&D waste in technical landfills to minimize the problems associated with transition zones and improve environmental behavior.

## 2. Materials and Methods

### 2.1. BB Sleeper Characterization

It is vital to acknowledge Infraestruturas de Portugal S.A. (IP) for providing and making possible the characterization of the state of conservation of the BB sleepers to obtain the parameters used in the simulations. The BB sleepers were obtained through the repair and rehabilitation of railway lines, removing them directly from the railway and storage in local storage centers, preserving the overall integrity of the elements. We chose BB sleepers such as C&D waste elements considering the technical railway requirements. After being received at the University Minho laboratory, the C&D waste elements were stored without having contact with the outside environment to maintain their state of conservation and to not damage their integrity. Then, a visual characterization was made, where the state of conservation was evaluated considering the normative requirements. In the characterization, parameters such as the length and width of the concrete and the joining steel element were measured. The material characterization parameters adopted in the simulations are stipulated and detailed in following the parameters of the case study model [10]. The spacing between the sleepers, which was implemented as the rail support, was 0.60 m between their axes, and the dimensions of the concrete of the sleepers were 0.20 m × 0.21 m cross-section with a length of 0.67 m. The steel profile connecting the two concrete blocks had an L shape of 60 mm × 60 mm and a thickness of 5.00 mm. The free distance between concrete blocks was 1.12 m, thus obtaining a total length of 2.46 m. A schematic representation of the studied BB is shown in Figure 1.

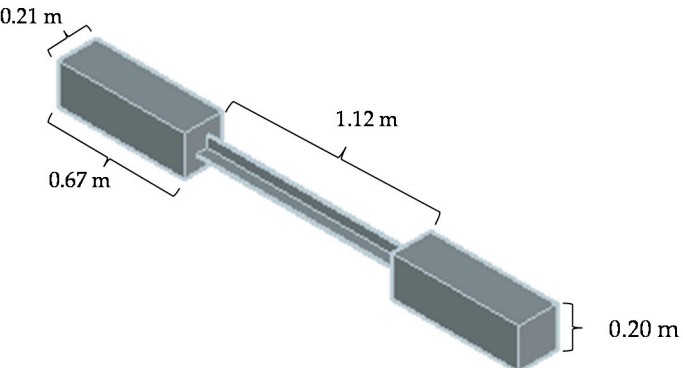

**Figure 1.** 3D drawing of the bi-block concrete sleepers.

### 2.2. Numerical Modelling

To create a structurally and environmentally feasible alternative that can reduce the existing problems in railways, specifically in the transition zones between the embankment and the structure, the characterization and analysis of different BB concrete sleeper arrangements as reinforcement elements were studied. Therefore, several solutions were determined and simulated in DIANA FEA software and SimaPro software to analyze the levels of displacements produced and to evaluate the environmental performance of each arrangement to achieve a technically feasible solution with the lowest environmental impact.

The finite element software DIANA FEA was implemented to perform the 3D simulation of the transition zone with the different implemented arrangements when applying a load of 100 kN in the direction from the embankment to the bridge, obtaining the response of the structure against the load applications along the length of the track, see Figure 2, for each solution presented. The elements and their respective properties that compose the model are detailed in Table 1. The models were verified, and the quality of the data obtained was validated by replicating a model where the behavior of transition zones treated with transition wedges was simulated [10], taking this paper as the base study model. The 3D simulation of the transition zone had the same load application and element properties of the base study, substituting the transition wedges to BB concrete sleeper arrangements as reinforcement elements with the aims of comparing the different solutions. In the simulation, hexa/quad and tetra/triangle mesh types of size 0.8 m were considered for all elements except the UIC60 rail in which a tetra/triangle mesh type of size 0.2 m was implemented.

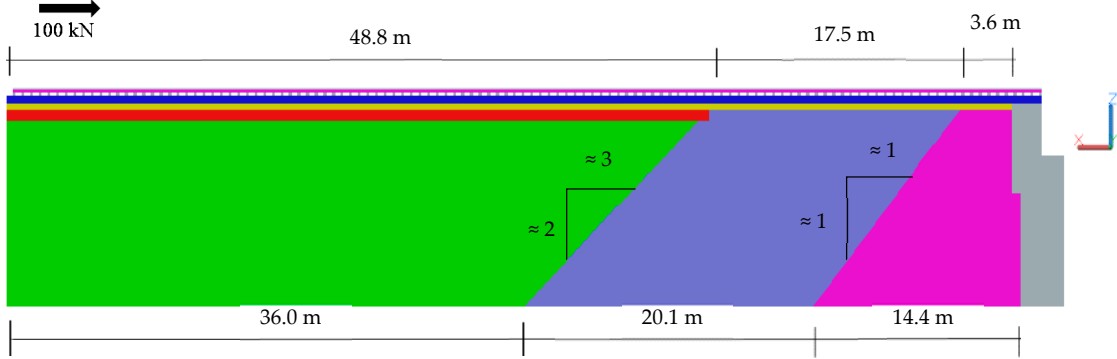

**Figure 2.** Longitudinal geometric arrangement and components with dimensions defined in the XZ plane for the base model with wedges and definitions of the mobile load and application direction.

**Table 1.** Characterization parameters of the materials adopted in the numerical simulations.

| Element | Parameter | Value | Unit |
|---|---|---|---|
| Sleeper (Bi-block in prestressed concrete)/Joint | Poisson ratio | 0.20 | [-] |
| | Young's modulus | 30 | [GPa] |
| | Density | 1700 | $[Kg/m^3]$ |
| Reinforcing steel | Poisson ratio | 0.30 | [-] |
| | Young's modulus | 200 | [GPa] |
| | Density | 7.85 | $[Kg/m^3]$ |
| UIC60 rail | Poisson ratio | 0.30 | [-] |
| | Young's modulus | 210 | [GPa] |
| | Density | 7800 | $[Kg/m^3]$ |
| Ballast | Thickness | 0.40 | [m] |
| | Poisson ratio | 0.10 | [-] |
| | Young's modulus | 200 | [MPa] |
| | Density | 1800 | $[Kg/m^3]$ |
| Sub-Ballast | Thickness | 0.30 | [m] |
| | Poisson ratio | 0.20 | [-] |
| | Young's modulus | 259.70 | [MPa] |
| | Density | 2200 | $[Kg/m^3]$ |
| Track-bed | Thickness | 0.60 | [m] |
| | Poisson ratio | 0.20 | [-] |
| | Young's modulus | 400 | [MPa] |
| | Density | 2200 | $[Kg/m^3]$ |
| Wedge 1 | Poisson ratio | 0.20 | [-] |
| | Young's modulus | 519.40 | [MPa] |
| | Density | 2150 | $[Kg/m^3]$ |
| Wedge 2 | Poisson ratio | 0.20 | [-] |
| | Young's modulus | 259.70 | [MPa] |
| | Density | 2080 | $[Kg/m^3]$ |
| Landfill | Poisson ratio | 0.20 | [-] |
| | Young's modulus | 60 | [MPa] |
| | Density | 1850 | $[Kg/m^3]$ |
| Foundation | Poisson ratio | 0.30 | [-] |
| | Young's modulus | 246 | [MPa] |
| | Density | 1850 | $[Kg/m^3]$ |

A recursive solution typology was implemented adopting a structure constituted by two inverted transition wedges (Figure 2) to obtain the behavior of the railway in the affected zones due to the vertical stiffness variation problems. The dimensions adopted in the 3D numerical simulations were obtained from the replica of the base study model [10] and are described below; they have a longitudinal dimension of 69.9 m at the rail level and 70.5 m at the foundation level. The colors implemented were chosen to recognize and differentiate the elements that compound the study case. Transversally, a measurement of 20 m was implemented, this section being considered as half of the track, thus allowing geometric and loading symmetry. The geometric arrangement adopted laterally aimed to simulate a surrounding foundation with a depth of 6.19 m. Vertically, the dimension from the base of the track-bed to the foundation was 11.1 m, which has the role of the track structure. Finally, the UIC60 rail was adopted as the axis and primary contact between the train and the railway due to its recurrent use in railways besides being based on the base case study model. The layouts of the ballast, sub-ballast, track-bed, embankment, and foundation layers are presented in Figure 3, in addition to the dimensions and components of the structure.

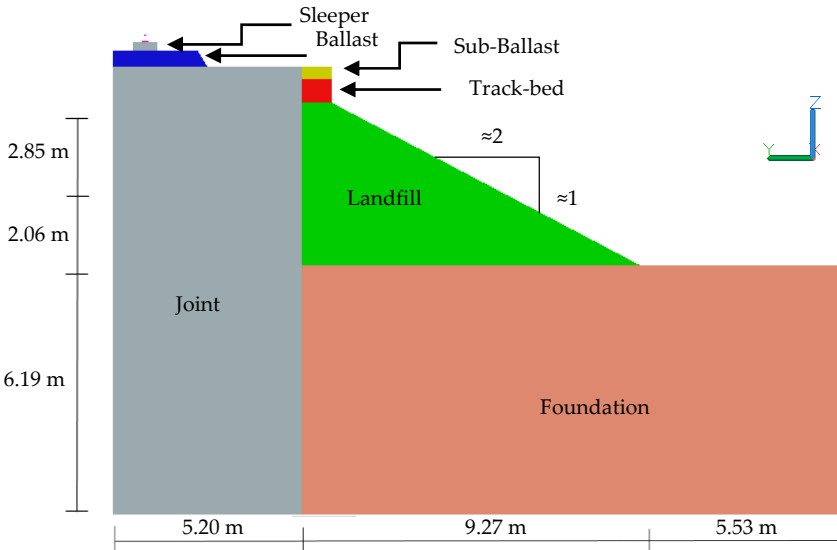

**Figure 3.** Transverse geometric arrangement and components with dimensions defined in the YZ plane viewed facing the joint.

A simulation without any type of reinforcement (NRM) was considered. The transition wedges were eliminated, and the BB sleepers were not implemented, being replaced by using a continuous foundation with a 6.19 m depth along the railway and immediately above the embankment layout.

Solutions that implement C&D waste, specifically BB sleepers as reinforcement elements, were developed to perform a comparison when implementing traditional solutions in contrast to innovative solutions that, depending on the arrangement along the length of the railway, simulate the response of conventional solutions by improving the stiffness properties of the embankment in the transition zones (Figure 4). The BB sleepers were grouped into two or three groups, following the type of arrangement, in such a way that the number of sleepers per row in each group along the length arranged by the transition wedges of the base model (BM) was kept constant. The studied arrangements are presented below, and parameters such as the horizontal distance (HD), vertically (VD), the number of sleepers (NS), and the levels implemented in each grouping were varied.

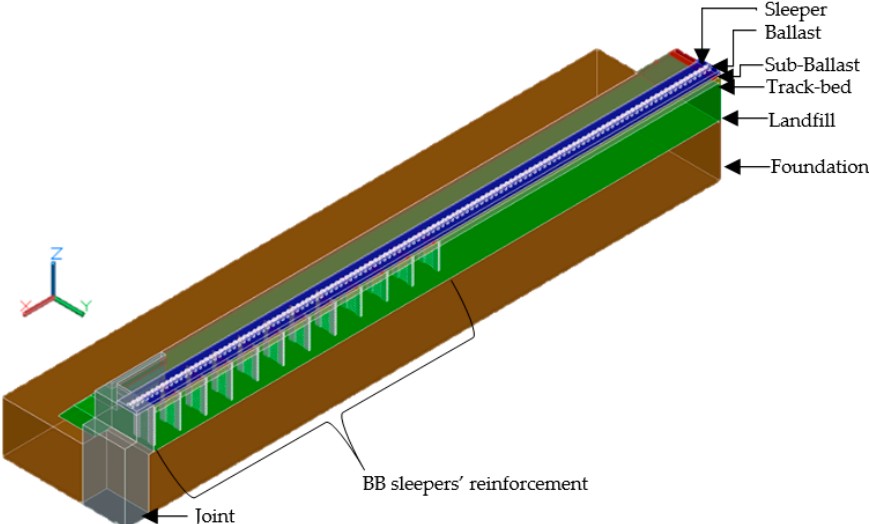

**Figure 4.** Horizontal BB sleeper reinforcement arrangement example.

The different types of arrangements were chosen due to adopting similar stiffness behavior responses in comparison to traditional solutions, such as transition wedges, and allowing the use of C&D waste constituting friendly environmental solutions.

### 2.2.1. Horizontal Arrangement (HA)

The reinforcement BB sleepers were arranged immediately below the ballast layer horizontally, i.e., perpendicular to the rail axis. The set with the highest NS and the one closest to the joint was set 1 with nineteen sleepers per row, set 2 had seventeen sleepers per row, and set 3 had fifteen sleepers per row, thus obtaining similar behaviors to the transition wedges. Variations in HD were performed from 0.8 m to 3.0 m by decreasing the number of rows but still maintaining the proportions of the two transition wedges with clusters 1 and 2. The purpose of cluster 3 was to perform a more graded transition (Figure 5).

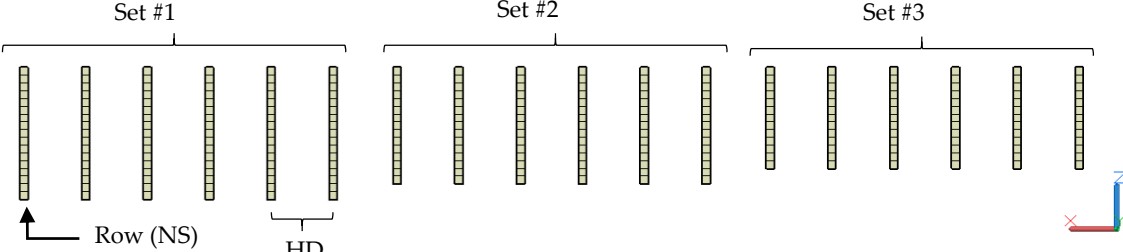

**Figure 5.** Definition of the various parameters in the arrangements of sleepers in the horizontal direction: Sets, HD, and row determination.

### 2.2.2. Vertical Arrangement (VA)

The reinforcement BB sleepers were arranged immediately below the ballast layer vertically. The cluster with the highest NS and the one closest to the joint was set 1 with eighteen sleepers per row, set 2 had twelve sleepers per row, and set 3 had six sleepers per row, maintaining a spacing of 1.5 m longitudinally between rows and the levels shown in Figure 6. The NS was varied for set 1 with six sleepers per row, set 2 with four sleepers per row, and set 3 with two sleepers per row. An NS extra variation was created to analyze the behavior response, implementing for set 1 twelve sleepers per row; for set 2, nine sleepers per row; and for set 3, six sleepers per row.

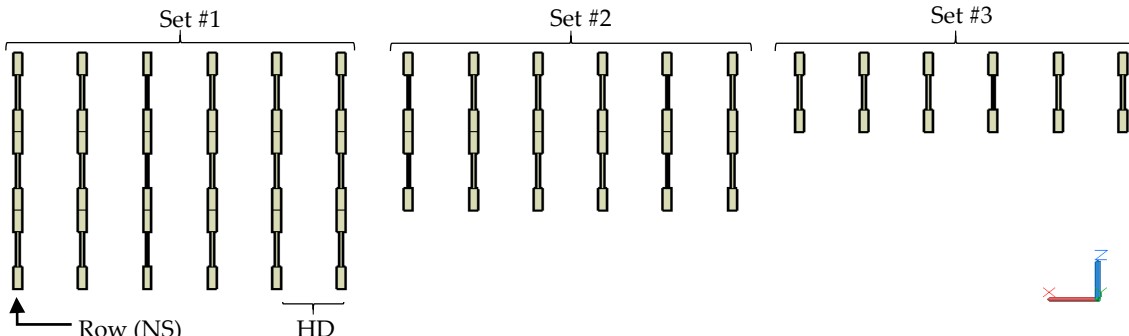

**Figure 6.** Definition of the various parameters in the arrangements of sleepers in the vertical direction: Sets, HD, and row determination.

### 2.2.3. Pyramidal Arrangement (PA)

Adopting a pyramidal arrangement of four levels (Figure 7), the reinforcement BB sleepers were arranged immediately below the ballast layer vertically to avoid the problem of the punching effect generated by the high loads and the reduced areas that transmit the loads to the ground through the reinforcement elements (Figure 6) to allow a more effective stress transition and greater contact surface.

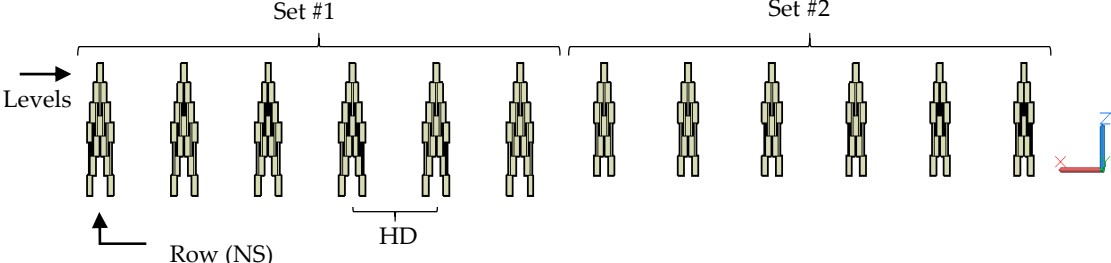

**Figure 7.** Definition of the various parameters in the pyramidal arrangement: Sets, HD, and row determination.

The set with the higher NS was the one closer to the joint. The row sets were formed by two BB sleeper lines per row in-depth, constituting set 1 with seven BB sleepers per line, fourteen per row, and set 2 with five BB sleepers per line, ten per row, thus obtaining similar behaviors to the transition wedges. Variations in HD were performed from 0.8 m to 3.0 m by decreasing the number of rows but still maintaining the proportions of the two transition wedges with clusters 1 and 2.

### 2.2.4. Single Horizontal Arrangement 90° (SHA90°)

The reinforcement BB sleepers were arranged immediately below the ballast layer horizontally and located in the middle of the BB sleepers that support the rail, i.e., parallel to the rail axis, thus implementing a single line of BB sleepers per row, Figure 8. The set with the higher NS and the one closer to the joint was set 1 with nineteen sleepers per row; set 2 had fifteen sleepers per row (SHA90° 19/15). The NS was varied for set 1 with twenty-five sleepers per row and for set 2 with twenty sleepers per row (SHA90° 25/20), thus obtaining similar behaviors in the transition wedges. The HD was varied from 1.0 m to 3.0 m by decreasing the number of rows but still maintaining the proportions of the two transition wedges with clusters 1 and 2.

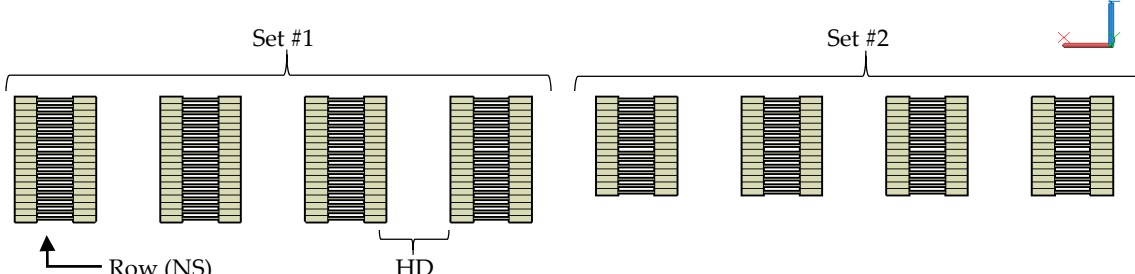

**Figure 8.** Definition of the parameters of variation in the arrangements of sleepers in the horizontal direction turned 90°: Sets, HD, and determination of rows.

### 2.2.5. Double Horizontal Arrangement 90° (DHA90°)

The reinforcement BB sleepers were arranged immediately below the ballast layer horizontally and were located in the middle of the BB sleepers that support the rail, i.e., parallel to the rail axis, thus implementing two lines of BB sleepers per row in-depth, Figure 8. The set with the higher NS and the one closer to the joint was set 1 with nineteen BB sleepers per line, thirty-eight per row; set 2 had fifteen BB sleepers per line, thirty per row (DHA90° 38/30). In addition, the NS was varied for set 1 with twenty-five BB sleepers per line, fifty per row, and set 2 with twenty sleepers per line, forty per row (DHA90° 50/40), thus obtaining similar behaviors in the transition wedges. Variations in HD were performed from 1.0 m to 3.0 m by decreasing the number of rows but still maintaining the proportions of the two transition wedges with sets 1 and 2.

## 3. Analysis of Numerical Simulation Results

This section will describe and discuss the response behavior of the different BB sleeper provisions assumed as reinforcement elements against a mobile punctual load assumed to be 100 kN along the railway.

### 3.1. Horizontal Arrangement (HA)

Following the observed response behavior, in Figure 9, it can be determined that adding the HD, creating a joint reinforcement system with shorter lengths between BB sleepers, substantially increased the vertical stiffness, and this was evidenced in the decrease in vertical displacements. Furthermore, large distances between the rows of reinforcing BB sleepers produced a non-joint reinforcement response, as evidenced by implementing 3.0 m distances, creating variations in the stiffness of the structure.

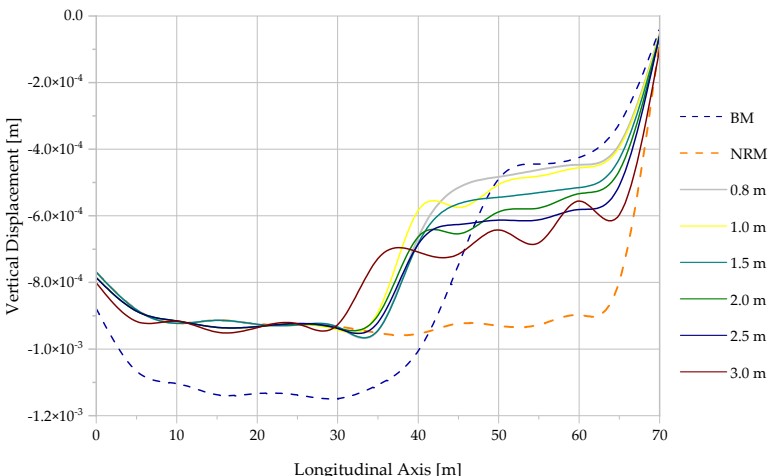

**Figure 9.** Vertical displacements when implementing the HA.

### 3.2. Vertical Arrangement (VA)

It was determined that adding the NS has no substantial influence on the embankment stiffness considering the results obtained in Figure 10, and therefore, the vertical displacements obtained did not resemble the behavior expected from traditional solutions. Thus, it was determined that the vertical arrangement is a structurally unfeasible alternative and should not be considered.

### 3.3. Pyramidal Arrangement (PA)

Following the response behavior by implementing the pyramidal arrangement (Figure 11), an increase in stiffness was observed when reducing the distance between rows of reinforcement BB sleepers, producing a decrease in vertical displacement. Similarly, with the horizontal arrangement, a response with more significant variation was also observed when relatively longer distances were presented. Moreover, when comparing the horizontal and vertical reinforcement NS per row, it was concluded that when implementing a pyramidal arrangement, a smaller number of sleepers is required, and higher stiffness values can be achieved for the structure under study. On the other hand, it was determined that implementing this type of arrangement on site can cause higher degrees of complexity due to the elevated weights and configurations of the BB sleepers.

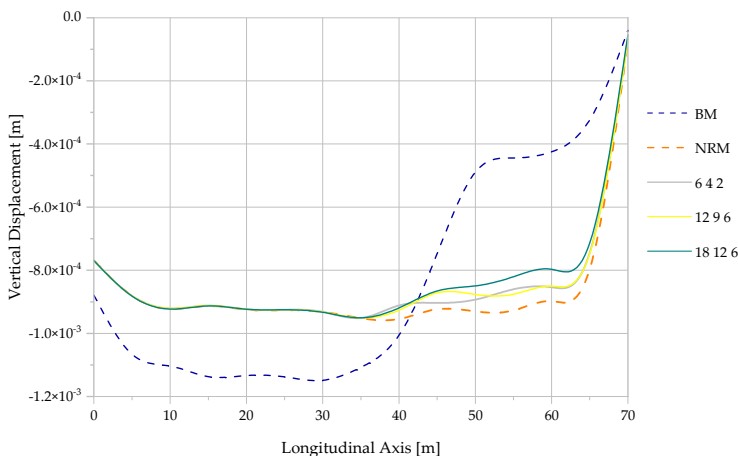

**Figure 10.** Vertical displacements when implementing the VA.

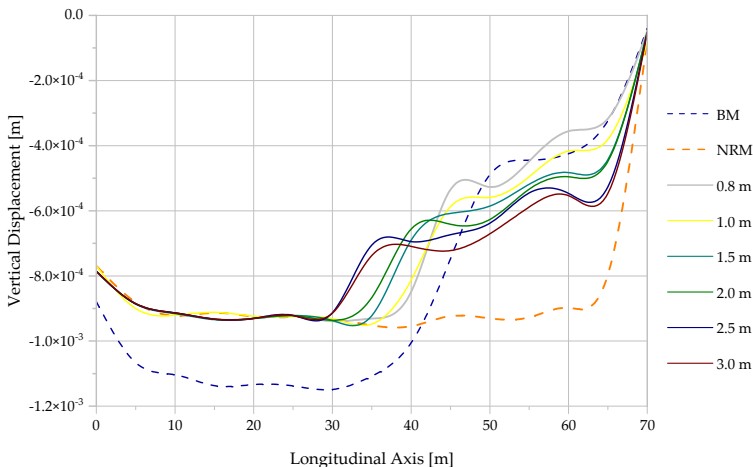

**Figure 11.** Vertical displacements when implementing the PA.

### 3.4. Single and Double Horizontal Arrangement 90°

Observing the data obtained when implementing the single and double horizontal 90° arrangement presented in Figures 12–15, a non-significant variation in vertical displacements was determined when increasing the horizontal distances between the rows of sleepers from 1.0 m to 3.0 m.

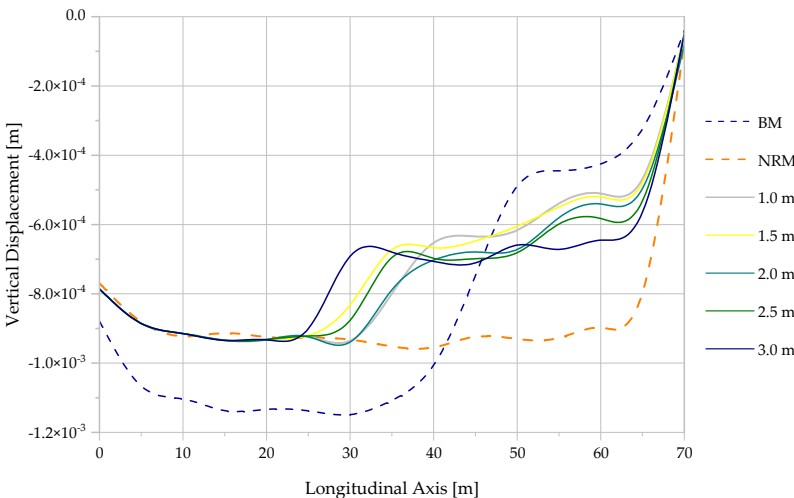

**Figure 12.** Vertical displacements when implementing the SHA90° with 19 and 15 BB sleepers per row in sets 1 and 2, respectively.

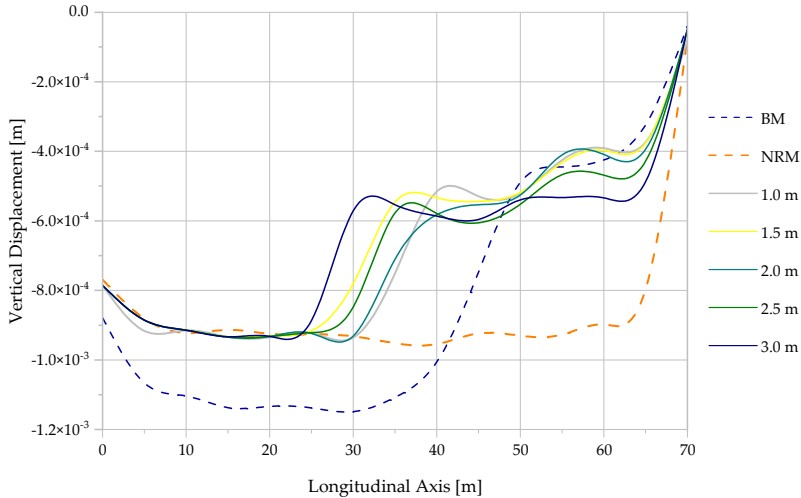

**Figure 13.** Vertical displacements when implementing the SHA90° with 25 and 20 BB sleepers per row in sets 1 and 2, respectively.

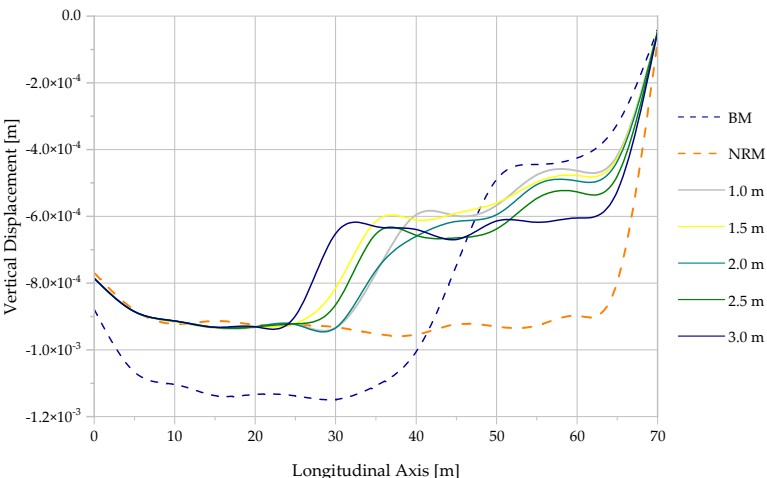

**Figure 14.** Vertical displacements when implementing the DHA90° with 38 and 30 BB sleepers per row in sets 1 and 2, respectively.

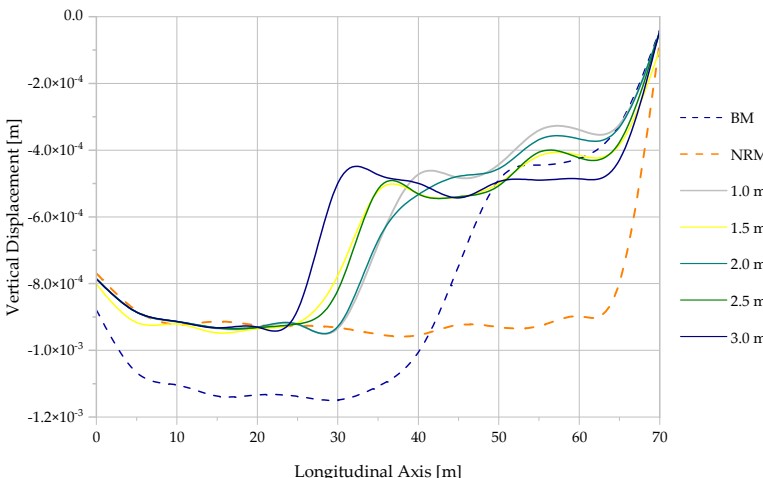

**Figure 15.** Vertical displacements when implementing the DHA90° with 50 and 40 BB sleepers per row in sets 1 and 2, respectively.

It can be seen that adding NS to each set, as evidenced in Figures 13 and 15, maintained the behavior trend but reduced the vertical displacements produced by the applied loads; this can occur because the BB sleepers implemented as reinforcement added a degree of stiffness that did not have a proportional effect.

Analysis of the vertical displacements obtained from numerical simulations indicated that the double horizontal arrangement of 90° presented in Figure 15, used in set 1 with twenty-five BB sleepers per line, fifty per row, and set 2 with twenty sleepers per line, forty per row, was the one with the smallest vertical displacements and therefore had the highest levels of stiffness and was determined as the best alternative solution in transition zones at the structural level.

## 4. Life Cycle Assessment (LCA)

The comparative analysis and the aggregation of indicators were developed using the Multicriteria Decision Support Methodology for the Relative Sustainability Assessment of Building Technologies (MARS-SC) considering only the environmental aspects. This method is conducted in five steps developed sequentially: (i) definition of sustainability indicators; (ii) quantification of indicators (including life cycle inventory); (iii) normalization of indicators; (iv) aggregation of indicators; and (v) calculation of the sustainable score and evaluation [17,18].

### 4.1. Declared Unit and System Boundaries

The boundaries mark the embodied "cradle-to-gate" environmental impacts of the different arrangements and layouts implemented, as well as the environmental impacts that result from the transport of materials to building solutions that smooth and mitigate stiffness variations, the complete construction of the structure, and its recurrent use for which it was designed. The declared unit depends on the objective of the life cycle analysis and therefore constitutes a reinforced transition zone. Figure 16 presents, in a simplified way, the processes included in the LCA analysis and the limits of the study, called boundaries. The presented system was adapted according to the provisions created for each scenario and, due to the different variants presented, the process represented with blue color is the one that considers the scenario where the transition zone is reinforced with BB sleepers.

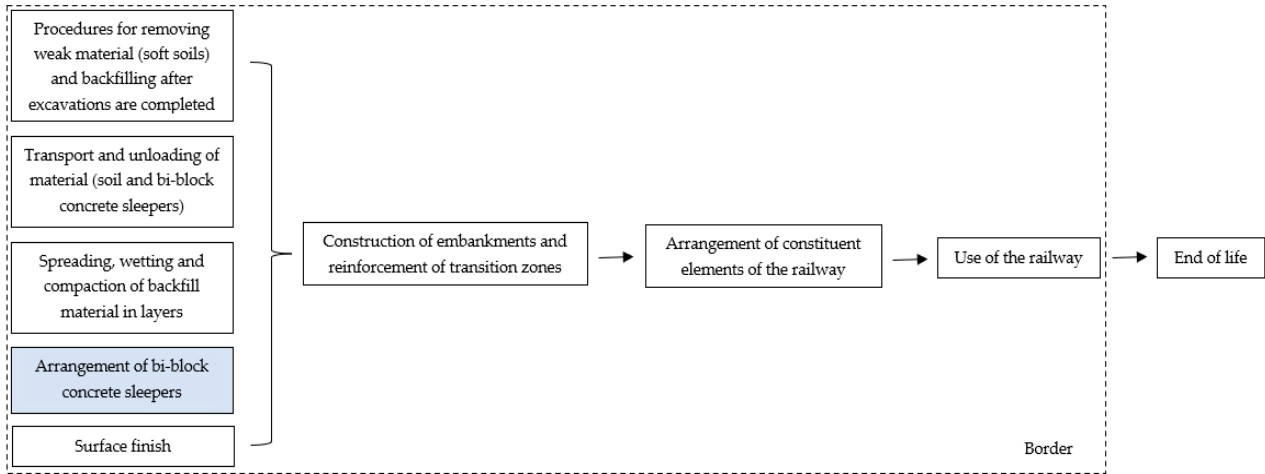

**Figure 16.** Processes considered in the environmental analysis of the different arrangements of BB sleepers and the study boundary.

*4.2. Inventory Analysis*

In the present study, raw material production and transport for each case study were included in the inventory. Tables 2 and 3 showed the inventory of materials and transport considered for each reinforcement arrangement, respectively. This inventory considered the own volumes of each disposal alternative. The life cycle analysis software SimaPro 7.3.3 was used to quantify the impact categories in a straightforward manner.

**Table 2.** Inventory of material inputs for each reinforcement arrangement.

| | NRM | BM | HA | PA | SHA90° 19/15 | SHA90° 25/20 | DHA90° 38/30 | DHA90° 50/40 | Units |
|---|---|---|---|---|---|---|---|---|---|
| **Ballast** | 71.04 | 71.04 | 71.04 | 71.04 | 71.04 | 71.04 | 71.04 | 71.04 | m$^3$ |
| **Sub-Ballast** | 125.24 | 125.24 | 125.24 | 125.24 | 125.24 | 125.24 | 125.24 | 125.24 | m$^3$ |
| **Track-bed** | 238.95 | 186.98 | 234.33 | 237.75 | 237.24 | 237.58 | 233.48 | 232.11 | m$^3$ |
| **Landfill** | 2530.43 | 2992.75 | 2508.88 | 2517.27 | 2522.45 | 2521.53 | 2520.39 | 2511.62 | m$^3$ |
| **Foundation** | 6733.48 | 4472.31 | 6733.48 | 6733.48 | 6733.48 | 6732.11 | 6733.48 | 6726.64 | m$^3$ |
| **Wedge 1** | N/A | 500.12 | N/A | N/A | N/A | N/A | N/A | N/A | m$^3$ |
| **Wedge 2** | N/A | 1051.94 | N/A | N/A | N/A | N/A | N/A | N/A | m$^3$ |
| **NS** | N/A | N/A | 459.00 | 252.00 | 170.00 | 180.00 | 272.00 | 450.00 | unit |

**Table 3.** Inventory of transport inputs for each reinforcement arrangement.

| | NRM | BM | HA | PA | SHA90° 19/15 | SHA90° 25/20 | DHA90° 38/30 | DHA90° 50/40 | Units |
|---|---|---|---|---|---|---|---|---|---|
| Ballast | 127.87 | 127.87 | 127.87 | 127.87 | 127.87 | 127.87 | 127.87 | 127.87 | tkm |
| Sub-Ballast | 275.53 | 275.53 | 275.53 | 275.53 | 275.53 | 275.53 | 275.53 | 275.53 | tkm |
| Track-bed | 525.69 | 411.35 | 515.53 | 523.04 | 521.93 | 522.68 | 513.65 | 510.64 | tkm |
| Landfill | 4681.29 | 5536.59 | 4641.43 | 4656.94 | 4666.52 | 4664.84 | 4662.73 | 4646.49 | tkm |
| Foundation | 12,456.94 | 8273.77 | 12,456.94 | 12,456.94 | 12,456.94 | 12,454.41 | 12,456.94 | 12,444.29 | tkm |
| Wedge 1 | N/A | 1075.26 | N/A | N/A | N/A | N/A | N/A | N/A | tkm |
| Wedge 2 | N/A | 2188.04 | N/A | N/A | N/A | N/A | N/A | N/A | tkm |
| Sleepers | N/A | N/A | 87.21 | 47.88 | 32.30 | 34.20 | 51.68 | 85.50 | tkm |

The specific consumption of raw materials, energy, fuels, and emissions released to air, water, and soil during the construction of the transition zone located in Portugal was considered. In addition, it was deemed that the backfill material and BB sleepers already arranged in storage sheds were located 10 km and 200 km, respectively.

SimaPro software was implemented to evaluate the potential environmental impacts of creating transition zones reinforced with BB sleepers. It is important to note that in

Portugal, BB sleepers removed from the railway track are considered waste and therefore do not have an economic value, so only the environmental impacts of their transport to the transition zone, and not those of their production, are considered. The information was collected from one of the most internationally accredited generic environmental databases of the Ecoinvent V2.2 report [20].

To quantify the environmental impacts generated by the different arrangement alternatives, we proceeded to the description of the processes taken into consideration for each constituent element of the proposed solution that was implemented in the software considering the available databases presented in Table 4.

In calculating the number of hours that should be used to create the scenarios considered a solution to the stiffness problems (yields), recommendations made by the GTR guide were implemented for each compacted layer thickness [21]. A moderate degree of compaction, similar thickness, compactor speed of approximately 5.0 km/h, 2.0 m wide mono compactor-type compaction machines, and materials classified as type A and C were assumed. In addition, the values of Q/L, which is considered the theoretical yield corresponding to a mono compactor, were calculated and used to calculate the yield for each soil layer assumed in the previously determined solutions. The yields and Q/L values for each layer conforming to the proposed solutions are presented in Table 5.

**Table 4.** Processes implemented to carry out the quantification of the environmental impact.

| Ecoinvent Processes | Ballast | Sub-Ballast | Track-Bed | Landfill | Foundation | Wedge 1 | Wedge 2 | Sleepers | Railroad |
|---|---|---|---|---|---|---|---|---|---|
| **Crushed stone 16/32 mm, open pit mining, production mix, at plant, undried RER S** | x | x | | | | x | x | | |
| **Gravel, round {CH} \| gravel and sand quarry operation \| Alloc Def, S** | | | x | | | | | | |
| **Clay and soil from quarry, EU27** | | | | x | x | | | | |
| **Cement, Portland {Europe without Switzerland} \| market for \| Alloc Def, S** | | | | | | x | | | |
| **Loader operation, large, NE-NC/RNA** | x | x | x | x | x | x | x | | |
| **Transport, combination truck, diesel powered/US** | x | x | x | x | x | x | x | x | |
| **Railway track {row} \| construction \| Alloc Def, S** | | | | | | | | | x |

**Table 5.** Calculated yields for each layer of material following the proposed arrangements.

| | Q/L [$m^3h*m$] | BSC [$m^3$] | Yield [$h*m$] | BM [$m^3$] | Yield [$h*m$] | 19/17/15 [$m^3$] | Yield [$h*m$] | 14/7 4N [$m^3$] | Yield [$h*m$] | 19/15 90° S [$m^3$] | Yield [$h*m$] | 25/20 90° S [$m^3$] | Yield [$h*m$] | 19/15 90° D [$m^3$] | Yield [$h*m$] | 25/20 90° D [$m^3$] | Yield [$h*m$] |
|---|---|---|---|---|---|---|---|---|---|---|---|---|---|---|---|---|---|
| Ballast | 180 | 71.04 | 0.39 | 71.04 | 0.39 | 71.04 | 0.39 | 71.04 | 0.39 | 71.04 | 0.39 | 71.04 | 0.39 | 71.04 | 0.39 | 71.04 | 0.39 |
| Sub-Ballast | 180 | 125.24 | 0.70 | 125.24 | 0.70 | 125.24 | 0.70 | 125.24 | 0.70 | 125.24 | 0.70 | 125.24 | 0.70 | 125.24 | 0.70 | 125.24 | 0.70 |
| Track-bed | 180 | 238.95 | 1.33 | 186.98 | 1.04 | 234.33 | 1.30 | 237.75 | 1.32 | 237.24 | 1.32 | 237.58 | 1.32 | 233.48 | 1.30 | 232.11 | 1.29 |
| Landfill | 200 | 2530.43 | 12.65 | 2992.75 | 14.96 | 2508.88 | 12.54 | 2517.27 | 12.59 | 2522.45 | 12.61 | 2521.53 | 12.61 | 2520.39 | 12.60 | 2511.62 | 12.56 |
| Foundation | 200 | 6733.48 | 33.67 | 4472.31 | 22.36 | 6733.48 | 33.67 | 6733.48 | 33.67 | 6733.48 | 33.67 | 6732.11 | 33.66 | 6733.48 | 33.67 | 6726.64 | 33.63 |
| Wedge 1 | 180 | N/A | N/A | 500.12 | 2.78 | N/A | N/A | N/A | N/A | N/A | N/A | N/A | N/A | N/A | N/A | N/A | N/A |
| Wedge 2 | 180 | N/A | N/A | 1051.94 | 5.84 | N/A | N/A | N/A | N/A | N/A | N/A | N/A | N/A | N/A | N/A | N/A | N/A |

*4.3. Impact Evaluation*

Life cycle inventory data were converted into a potential environmental impact using life cycle impact assessment (LCIA) methods. In the MARS-SC methodology, the environmental performance assessment is based on the environmental impact categories evidenced in Table 6, adapted from [14]. Although MARS-SC does not consider the depletion of abiotic soil and water resources, this indicator was considered in this study using a similar approach to the one used in studies with the same goal [22–25].

**Table 6.** Indicators, units, and methods of quantification.

| Environmental Indicators | Units | LCIA Methods |
| --- | --- | --- |
| Global warming (GWP 100) | [Kg $CO_2$ eq] | CML-IA baseline V3.04/EU25+3, 2000 |
| Ozone depletion (ODP) | [Kg CFC − 11 eq] | CML-IA baseline V3.04/EU25+3, 2000 |
| Potential acidification (AP) | [Kg $SO_2$ eq] | CML-IA baseline V3.04/EU25+3, 2000 |
| Potential eutrophication (EP) | [Kg $PO_4$ eq] | CML-IA baseline V3.04/EU25+3, 2000 |
| Photochemical ozone creation (POCP) | [Kg $C_2H_4$ eq] | CML-IA baseline V3.04/EU25+3, 2000 |
| Abiotic depletion potential of fossil resources (ADP_FF) | [MJ eq] | Cumulative energy demand V1.08 |
| Depletion of abiotic soil and water resources (ERA) | [Kg Sb eq] | CML-IA baseline V3.04/EU25+3, 2000 |

*4.4. Normalization*

To avoid scale effects in the aggregation of the parameters of the different indicators and to minimize the possibility that some of the parameters are not interpreted in the best and correct way, it is necessary to normalize the indicators [18]. The normalization was done using the Diaz–Balteiro equation [26] shown below.

$$\overline{P_i} = \frac{P_i - P_{*i}}{P_i^* - P_{*i}} \quad \forall i \tag{1}$$

In this equation, $P_i$ is the value of the parameter *i*. $P_i^*$ and $P_{*i}$ are the respective best and worst values of the sustainability parameter of *i* among the products analyzed. The normalization converts the values in a scale delimited between 0 (worst value) and 1 (best value) and transforms the value of each dimensioned indicator [18].

*4.5. Aggregation and Global Assessment*

The following equation calculates the aggregation of each environmental indicator in terms of an overall indicator, describing the overall environmental performance ($ND_A$).

$$ND_A = \sum_{i=1}^{n} w_i * \overline{P_i} \tag{2}$$

The overall indicator ($ND_A$) is the average weighting for each standardized indicator $P_i$ while $w_i$ is the contribution of indicator *i* to the overall environmental performance. The sum of all weights must equal 1.0 [18]. In addition, this study considers the weights defined in a study developed by the Science Advisory Board (SAB) of the US Environmental Protection Agency [19] which correspond as follows: (i) GWP 38%, (ii) ODP 12%; (iii) AP 12%; (iv) EP 12%; (v) POCP 14%; (vi) ADP_FF 12%.

**5. Analysis of Life Cycle Assessment Results**

The results obtained in the life cycle assessment begin with the comparison of the different alternatives generated by the different provisions to observe the percentage impact of each variant previously studied in the reinforcement of the transition zones. These results are presented in Table 7, where reference is made to each sustainability indicator.

**Table 7.** Summary impact of sustainability indicators on the distinctive form arrangements.

| Impact Category | Unit | BM | PA | HA | SHA90° 19/15 | DHA90° 38/30 | SHA90° 25/20 | DHA90° 50/40 |
|---|---|---|---|---|---|---|---|---|
| GWP 100 | [kg $CO_2$ eq] | $2.75 \times 10^5$ | $2.44 \times 10^5$ | $2.44 \times 10^5$ | $2.44 \times 10^5$ | $2.44 \times 10^5$ | $2.44 \times 10^5$ | $2.44 \times 10^5$ |
| ODP | [kg CFC-11 eq] | $1.23 \times 10^{-2}$ | $2.21 \times 10^{-3}$ | $2.21 \times 10^{-3}$ | $2.21 \times 10^{-3}$ | $2.21 \times 10^{-3}$ | $2.21 \times 10^{-3}$ | $2.21 \times 10^{-3}$ |
| AP | [kg $SO_2$ eq] | $1.31 \times 10^3$ | $1.10 \times 10^3$ | $1.10 \times 10^3$ | $1.10 \times 10^3$ | $1.10 \times 10^3$ | $1.10 \times 10^3$ | $1.10 \times 10^3$ |
| EP | [kg $PO_4$ eq] | $1.49 \times 10^2$ | $1.30 \times 10^2$ | $1.30 \times 10^2$ | $1.30 \times 10^2$ | $1.30 \times 10^2$ | $1.30 \times 10^2$ | $1.30 \times 10^2$ |
| POCP | [kg $C_2H_4$ eq] | $8.95 \times 10^1$ | $8.45 \times 10^1$ | $8.45 \times 10^1$ | $8.46 \times 10^1$ | $8.46 \times 10^1$ | $8.46 \times 10^1$ | $8.46 \times 10^1$ |
| ADP_FF | [MJ eq] | $3.28 \times 10^6$ | $3.17 \times 10^6$ | $3.17 \times 10^6$ | $3.17 \times 10^6$ | $3.17 \times 10^6$ | $3.17 \times 10^6$ | $3.17 \times 10^6$ |
| ERA | [kg Sb eq] | $1.37 \times 10^{-1}$ | $1.45 \times 10^{-1}$ | $1.45 \times 10^{-1}$ | $1.45 \times 10^{-1}$ | $1.45 \times 10^{-1}$ | $1.45 \times 10^{-1}$ | $1.45 \times 10^{-1}$ |

Taking into consideration the environmental impact of traditional and non-traditional solutions in Table 7, one can determine a higher ozone depletion in the recurring solutions, i.e., when implementing the transition wedges as a solution alternative, thus producing an increase in the amount of ultraviolet radiation that passes through to the earth's surface. However, when analyzing the solutions implementing RCD, it can be determined that they present lower and similar values to each other. Therefore, it is determined that the use of waste as an alternative reinforcement directly affects the environmental impacts, reducing the environmental effects due to the construction sector.

Based on Table 7, it can be concluded that by implementing a large amount of BB sleepers, a decrease in the percentage of photochemical ozone production occurs, safeguarding human health, ecosystems, and agriculture, thus obtaining considerably lower values. The results obtained in the life cycle assessment (Table 7) were normalized to analyze and compare the environmental impact according to Equation (1). The normalized results are shown in Table 8 for each arrangement. Furthermore, based on a comparison of the normalized data (Table 8), the construction of transition wedges as structures to solve problems such as variations in the vertical stiffness of the track, displacements, and rotations of the bridge deck, as well as differential settlements in the transition zones, affects and damages the environment considerably, thus generating more significant environmental impacts and preventing this from being the most sustainable alternative. In contrast, BB sleepers are renowned as alternatives that can reduce environmental impacts, solving problems of stiffness variation and differential settlements in transition zones. The BB sleepers, based on the results obtained, are environmentally friendly solution alternatives and, regardless of the adopted layout, are more sustainable.

**Table 8.** Summarized standardized impact of sustainability indicators in the distinctive form arrangements.

| Impact Category | BM | PA | HA | SHA90° 19/15 | DHA90° 38/30 | SHA90° 25/20 | DHA90° 50/40 |
|---|---|---|---|---|---|---|---|
| GWP 100 | 0 | 0.989 | 0.996 | 0.985 | 0.988 | 0.987 | 1.000 |
| ODP | 0 | 0.999 | 1.000 | 0.999 | 1.000 | 0.999 | 1.000 |
| AP | 0 | 0.993 | 0.997 | 0.990 | 0.992 | 0.991 | 1.000 |
| EP | 0 | 0.989 | 0.996 | 0.986 | 0.990 | 0.987 | 1.000 |
| POCP | 0 | 0.977 | 0.992 | 0.969 | 0.973 | 0.973 | 1.000 |
| ADP_FF | 0 | 0.960 | 0.987 | 0.946 | 0.955 | 0.952 | 1.000 |

The use of non-traditional solutions with the implementation of BB sleepers as reinforcement RCD led to a decrease in environmental impacts when compared to current solutions such as transition wedges, regardless of the percentage of raw material substitution, i.e., soil implemented for each layer of the landfill and the amount of RCD. The results also showed that the values for the provisions with a higher amount of BB sleepers presented a better alternative in a way that is more efficient from a technical, economic, and environmental point of view. At this stage, it is necessary to highlight the effect of the attribution step of the results obtained.

Figure 17 presents the sustainability profiles and the overall environmental performances (ND$_A$) of each reinforcement arrangement under study. In the sustainability profiles, the drawn area represents the performance of each adopted arrangement. At the level of each impact category, the best shape arrangement when implementing BB sleepers as reinforcement elements is the one with a value closest to one. It was found that the DHA90° 50/40 arrangement presented the best overall environmental performance (ND$_A$ = 1.00), and in traditional solutions, such as in this study, the implementation of transition wedges (BM) presented the worst performance (ND$_A$ = 0.00).

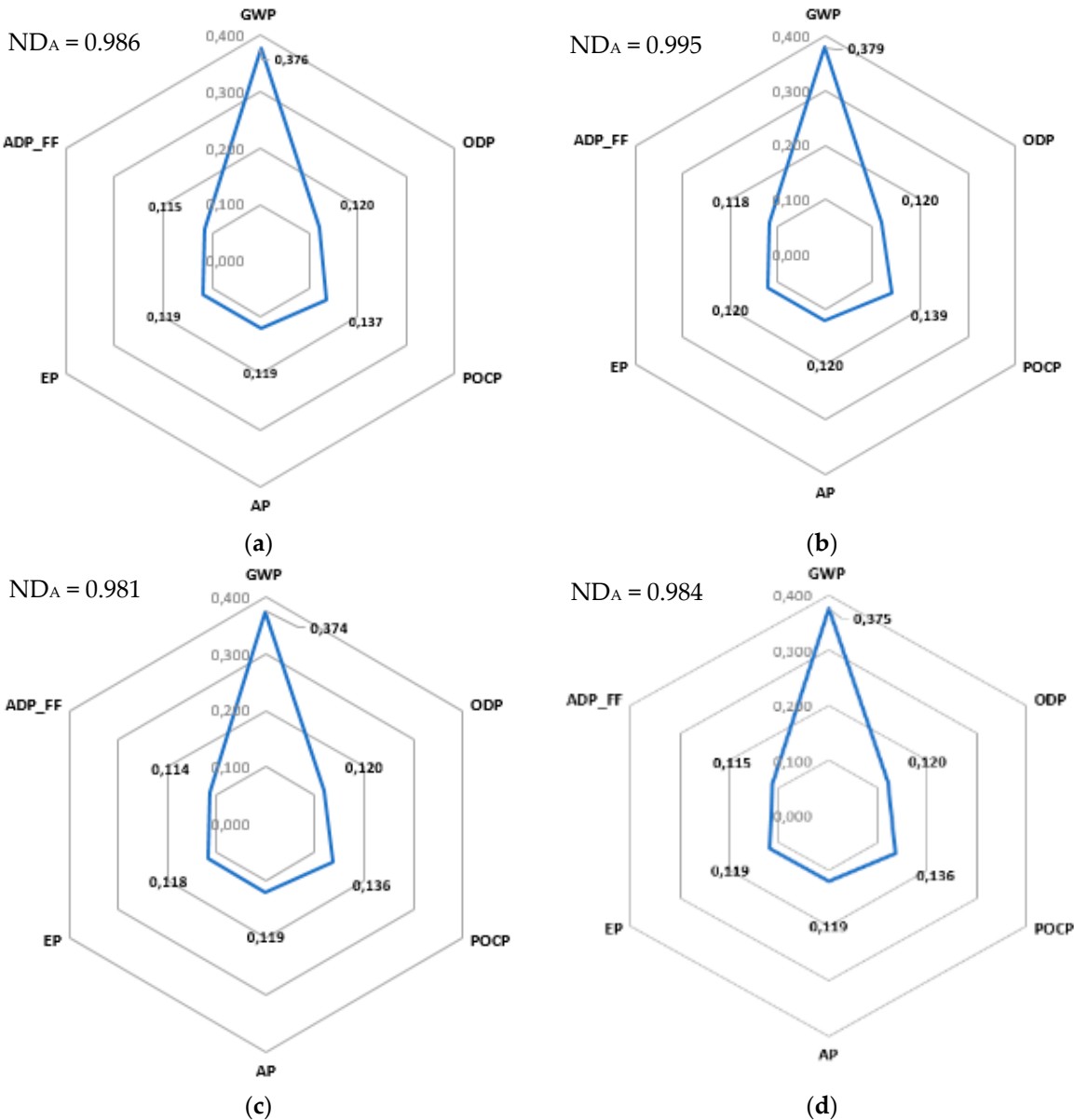

**Figure 17.** *Cont.*

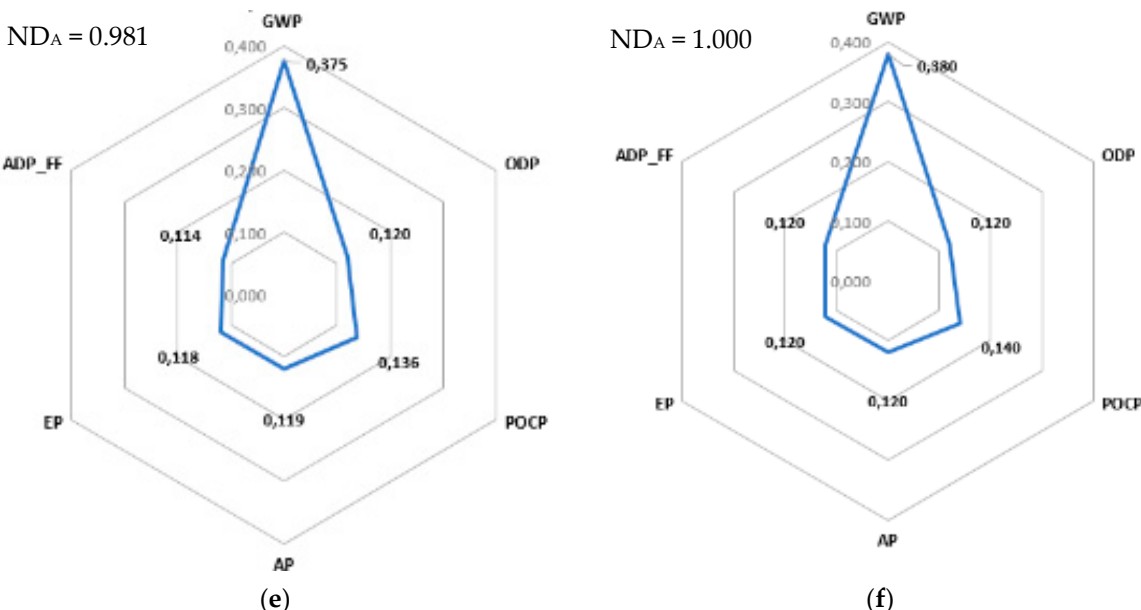

**Figure 17.** Sustainability profiles and environmental performance values of each form layout and respective overall environmental performance (ND$_A$). Legend: (**a**) PA; (**b**) HA; (**c**) SHA90° 19/15; (**d**) SHA90° 38/30; (**e**) DHA90° 25/20; (**f**) DHA90° 50/40.

Based on the results, it is possible to conclude that the use of BB sleepers as C&D waste significantly increases the environmental performance. Moreover, the use of these materials contributes to better compatibility between the construction sector and the sustainable development goals.

The results of the sensitivity analysis (Figure 17) show that, regardless of the provision adopted to increase the gradual stiffness levels of the transition zone, the environmental impact results will remain similar. This conclusion is underlined by the fact that the environmental performance values are similar, and their variation is almost zero. As a result, the impact attributed to the implementation of BB sleepers is lower than that of traditional alternatives used in construction. In addition, it should be taken into account that the environmental performance of the BM arrangements was zero, and therefore the sustainability profiles could not be drawn.

## 6. Final Remarks and Conclusions

The parameter of the vertical distance between Bi-Block (BB) concrete sleepers was determined as one of the possible factors that can influence the decrease in stiffness, i.e., the high transfer of effort of a material with high stiffness (e.g., BB sleepers) compared with material with significantly lower stiffness (e.g., soils that compose the layers of the embankment). Therefore, it is not recommended to implement spacing between the reinforcement elements to avoid this type of effect. Furthermore, changing the horizontal distance (HD) parameter between BB sleepers showed that implementing a horizontal spacing greater than 3.0 m affected the behavior of the controlled stiffness transition, and therefore, it was not possible to recreate the behavior of traditional solutions such as transition or inverted wedges.

By adopting different reinforcement arrangements, it was possible to conclude that implementing a vertical arrangement of BB sleepers did not obtain the expected degree of stiffness. The possible factor influencing the level of stiffness of the embankment is the punching effect, which can be caused by the transfer of loads from a material with high stiffness relative to material with low stiffness in a small surface area. Therefore, it is recommended to implement non-typical solutions (e.g., pyramidal arrangement) for a more effective load transition and to have a larger contact surface in order to solve problems associated with high load transfer in small areas. It is worth noting the recommendation not

to implement geometries that generate problems during construction due to the increased complexity and skilled labor required.

It was found that by implementing a horizontal arrangement parallel to the track axis, a high gradual stiffness increasing behavior was obtained in the treated transition zone. Therefore, this avoids the implementation of large amounts of sleepers, thus reducing the cost of material transport. However, it was observed that solutions classified as single or double did not show considerable differences in the degree of stiffness.

The sensitivity analysis enabled us to demonstrate that, regardless of the arrangement adopted to increase the gradual stiffness levels of the transition zone, the results remained similar. The highest stiffness levels were observed in the Double Horizontal Arrangement of 90° with fifty and forty sleepers per row in sets 1 and 2, respectively (DHA90° 50/40). In addition, it was determined that this is the most sustainable and therefore environmentally friendly alternative by having a better environmental performance. The use of Construction and Demolition (C&D) waste substitutes reduces the need to implement large quantities of necessary soil that, due to their extraction, transportation, and disposal process, generate more significant environmental impacts and reduce the pollution produced by the transportation and disposal of waste in sanitary landfills. As a result, the effect attributed to the implementation of C&D waste is lower when compared to traditional alternatives used in construction such as transition wedges, thus obtaining a better environmental performance and making it more sustainable.

The environmental impacts generated by the different formwork arrangements adopted makes it possible to conclude that the implementation of BB sleepers, as reinforcement elements in areas susceptible to stiffness variations, significantly decreases environmental impacts in addition to generating economic benefits due to the possibility of implementing elements that have no value in the construction market. It is noteworthy that the potential environmental impacts decrease with the increase in the substitution of extracted matter by BB sleepers, regardless of the arrangement adopted and, the high emission of $CO_2$ results from the high amounts of extraction and transport of raw material that occur during the construction of the transition zone.

These results allow us to conclude that, with the use of BB sleepers as C&D waste, it is possible to perform the construction of a transition zone with lower environmental impacts, maintaining an adequate stiffness transition that can solve problems of stiffness variations, in addition to having a more sustainable development and environmental contribution compared to the implementation of natural materials.

**Author Contributions:** Conceptualization, C.M., E.T. and J.T.; methodology, C.M. and E.T.; software, C.M.; validation, C.M., J.C.e.M. and R.M.; formal analysis, C.M.; investigation, C.M., E.T. and J.T.; resources, E.T. and J.T.; writing—original draft preparation, C.M.; writing—review and editing, E.T., J.T., R.M. and J.C.e.M.; visualization, C.M.; supervision E.T., J.T., R.M. and J.C.e.M.; project administration, E.T. and J.T.; funding acquisition, E.T., J.T., R.M. and J.C.e.M. All authors have read and agreed to the published version of the manuscript.

**Funding:** This work was partly financed by FCT/MCTES through national funds (PIDDAC) under the R&D Unit Institute for Sustainability and Innovation in Structural Engineering (ISISE), under reference UIDB/04029/2020. Also, was cofinanced by the Interreg Atlantic Area Programme through the European Regional Development Fund under SIRMA project (Grant No. EAPA_826/2018). This project has received funding from the European Union's Horizon 2020 research and innovation programme under grant agreement No. 769255. The sole responsibility for the content of this publication lies with the author.

**Institutional Review Board Statement:** Not applicable.

**Informed Consent Statement:** Not applicable.

**Data Availability Statement:** Not applicable.

**Acknowledgments:** The authors would like to thank ISISE—Institute for Sustainability and Innovation in Structural Engineering (PEst-C/ECI/UI4029/2011 FCOM-01-0124-FEDER-022681). The authors wish to thank IP—Infraestruturas de Portugal for supplying all materials used for this study. The sole responsibility for the content of this publication lies with the authors. It does not necessarily reflect the opinion of the European Union. Neither the Innovation and Networks Executive Agency (INEA) nor the European Commission are responsible for any use that may be made of the information contained therein.

**Conflicts of Interest:** The authors declare no conflict of interest.

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
