# Peer review of "Viability Study of the Application of Bi-Block Concrete Sleepers as a Solution for Technical Landfills"

_applsci, doi:10.3390/app12063065_

Round 1

Reviewer 1 Report

Overall, the manuscript was able to demonstrate the impacts of using concrete BB sleepers from demolition wastes to help solve issues related to stiffness variation in landfills as an alternative option to disposal. However, parts of the manuscript lack explanation and the following comments need to be addressed:

  1. The introduction does not provide sufficient basis for conducting the study. Additional literatures should be added in the Introduction section to answer the following questions:
    - What is the status of C&D wastes (e.g., quantities, impacts, common treatment options, policies, and regulations)?
    - How much concrete BB sleepers ended up in landfills compared to other C&D wastes that meet the technical standards? What level of environmental and economic impacts do these materials contribute? 
    - Describe the main issues specifically related to stiffness variations around landfills and their significance.
  2. Table 8 is mislabeled as Table 1.
  3. There should be some discussions of whether additional processing is needed to effectively use the BB sleepers. Are they taken directly from demolition sites to storage? Is the transportation from demolition sites to storage included in the system boundary? Are these wastes processed before application?

Reviewer 2 Report

Tabel 1 - The term "poisson ratio" is used rather than "p. coefficient"; in the chapter 2.1 only BB Sleeper is described - and the rest, eg. Ballast what is it, etc.?; how (where from) were the values in table 1 (e.g. for landfill) determined (taken)?

Line 93 - why was the value of 100kN chosen?

lines 108 and 109 - the dimensions given do not agree with those resulting from Figure 3.

Fig. 2 and 3 - what do the different colors mean? What are the dimensions in Figure 2 ?; why two drawings (no. 2 and 3) if in figure 3 - everything is what is shown in figure 2 (and more) - only the description "100kN" is missing!
Delete fig. 2, complete fig. 3.

fig. 4 - what does "Ballas" mean?

fig. 4 and descriptions in chapters 2.2.1-2.2.5 - where are these BB Sleepers reinforcment? If on the surface - they will not fit, e.g. VA set # 1 - has a length of 3x2.46 = 7.38, it is less than 5.20m! Where is the roadway located (line 132 - w)?

line 156-158 - I don't understand how they are arranged (one on top of another?).

line 164 - probably "PA"

line 174 - as shown in Fig. 11 - up to 3.0m!

line 205 - what was the load function (traffic)?

Fig. 10 - variant "12 9 6" - it is missing in the description in chapter 2.2.2.

lines 303-305 and 392-393 - the same information ...

lines 339 "e" - in Portuguese (must be "and").

lines 383 - Table 1? I think No. 7.

table 6, 7 and 8 - markings do not match (should be AP or PA? etc.)

Tables 7 and 8 show the values calculated according to equation 1 and 2, or any other? If so - which ones in what table?

Figure 17 - What are these values? (how they were calculated) do not correspond to the values given in tables (7, 8 and others). In fact, in individual points a-f there are no special differences in shape ... nothing can be seen from these figures.

Reviewer 3 Report

It is necessary to review the terminology used in the manuscript (e.g., LCA is "life", not "live" as in line 54).

The grammar is good overall, but good proofreading is necessary.

There needs to be consistent use of the decimal point or comma throughout the manuscript. 

The text should be reformatted in Figure 17 in order for the reader to accurately read the values without zooming in to >150%.

It is not articulated why the end-of-life step is not considered in the LCA.  If not considered, then it is not a complete LCA.

Round 2

Reviewer 1 Report

Authors have addressed my comments.

Reviewer 2 Report

The authors took into account all the comments - and made corrections in the text - I recommend the article to be accepted.